# ReCorD: Reasoning and Correcting Diffusion for HOI Generation

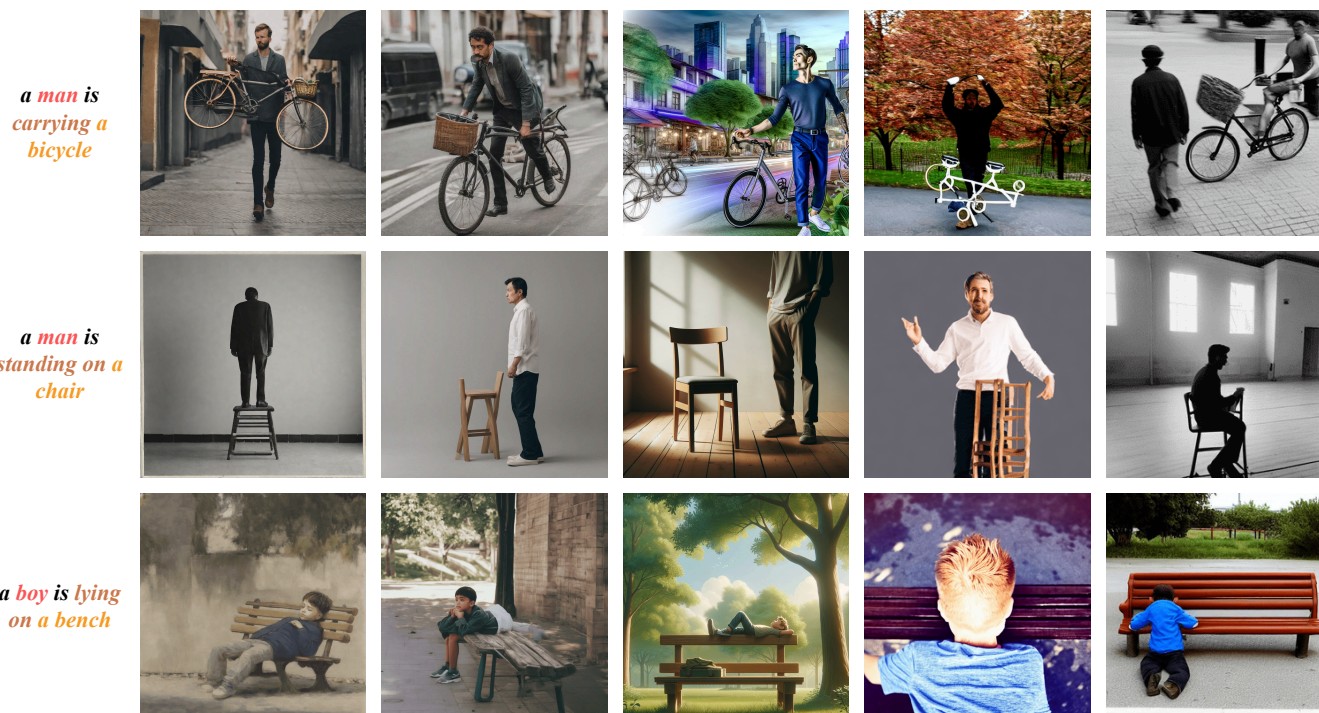

**Figure 1: Existing approaches frequently encounter difficulties in accurately interpreting prompts related to human-object interactions, resulting in misplaced objects and inaccurate poses. In contrast, our ReCorD shows significantly improved generation capabilities in diverse scenes, indicating superior proficiency in rendering complex interactions. Best view in colors as we use different colors to represent the triplets <human,object,interaction>.**

## ABSTRACT

Diffusion models revolutionize image generation by leveraging natural language to guide the creation of multimedia content. Despite significant advancements in such generative models, challenges persist in depicting detailed human-object interactions, especially regarding pose and object placement accuracy. We introduce a training-free method named *Reasoning and Correcting Diffusion (ReCorD)* to address these challenges. Our model couples Latent Diffusion Models with Visual Language Models to refine the generation process, ensuring precise depictions of HOIs. We propose an interaction-aware reasoning module to improve the interpretation of the interaction, along with an interaction correcting module to refine the output image for more precise HOI generation delicately. Through a meticulous process of pose selection and object positioning, ReCorD achieves superior fidelity in generated images while efficiently reducing computational requirements. We conduct comprehensive experiments on three benchmarks to demonstrate the significant progress in solving text-to-image generation tasks, showcasing ReCorD's ability to render complex interactions accurately by outperforming existing methods in HOI classification score, as well as FID and Verb CLIP-Score.

## CCS CONCEPTS

• **Information systems** → **Multimedia content creation**; • **Computing methodologies** → **Image manipulation**.

## KEYWORDS

multimodal image generation, visual language model, diffusion model, human-object interaction

**Unpublished working draft. Not for distribution.**

# 1 INTRODUCTION

In recent years, diffusion models have become a cornerstone in the field of multimedia processing, demonstrating remarkable success across a wide array of generative tasks [26, 35, 53, 61, 63, 71]. Among them, text-to-image (T2I) generation has attracted significant attention [8, 12, 40, 54, 62] due to its user-friendly nature leveraging natural language guidance. While state-of-the-art (SOTA) T2I diffusion models, such as SDXL [42] and DALL-E 3, have greatly improved image realism and expanded the conceptual possibilities of generation, ongoing challenges persist. In particular, they often encounter difficulties with text prompts that contain intricate human-object interactions (HOI) [23].

As illustrated in Figure 1, despite their training on extensive datasets, T2I methods like SDXL and DALL-E 3 exhibit flaws in rendering human poses or object placements. In the example of "a boy is lying on a bench", although DALL-E 3 precisely captures the lying pose, it incorrectly locates the boy, whereas SDXL places the boy on the bench accurately but does not manage to render the lying down pose. These inaccuracies may stem from the inherent biases or assumptions about the interaction between the given human and object, embedded within the large-scale datasets [50] used to train T2I models [16]. Such biases can lead to hallucination problems [34], resulting in models failing to generate images matching the intended interactions accurately. For instance, given the prompt "a man is carrying a bicycle" as in Figure 1, SDXL and DALL-E 3 might err in posture and object placement because the most common association in the datasets is "riding," leading to inaccuracies in depicting the intended interaction.

To enhance the accuracy of T2I models in generating interactions, a possible avenue is to ensure the correct positioning of both the human and the object within the image. Previously, several layout-to-image (L2I) models [13, 27, 29, 59, 69] have been proposed to include the layout of each object as additional input for diffusion models, aiming to gain more precise control over the output images. For instance, GLIGEN [29] retrain the model with the layout-annotated dataset by the Gated Self-Attention. DenseDiffusion [27] and BoxDiff [59] exemplify training-free L2I methods that necessitate supplementary inputs for operation. However, the requirement for user-specific layouts can be time-consuming and inconvenient for users. In addition, especially in scenarios involving HOI, simplistic inputs like boxes prove inadequate in capturing complex attributes such as posture and body orientation, which are crucial for the accurate depiction of interaction, ultimately resulting in suboptimal images. The deficiency is evident in Figure 1, where BoxDiff struggles to generate realistic interactions despite being provided with layout information as additional input.

On the other hand, alternative approaches have integrated Large Language Models (LLMs) to augment diffusion models, aiming better to grasp the nuances of textual prompts in image generation [15, 30, 43, 58, 60, 70]. Innovations such as LMD [30] and LayoutLLM-T2I [43] have pioneered to employ LLMs for creating more intuitive and accurate image outputs. LMD utilizes a dual-phase approach, initially using a pre-trained LLM to create a scene layout with captioned bounding boxes. It further proceeds with a layout-grounded controller to guide diffusion models. Additionally, LayoutLLM-T2I starts by generating a coarse layout and then implements a specially trained transformer module within the denoising UNet for fine-grained generations. Despite the progress made with LLM-assisted methods in image generation, a pivotal limitation arises when handling HOI. Predominantly dependent on textual prompts, these methods exhibit two possible shortcomings. Firstly, they may overlook the intricate spatial dynamics and nuanced interactions within an image due to the limited information the prompt provides. Secondly, in some cases, the LLMs may over-analyze the textual prompt, leading to hallucinations where they generate fabricated content that is not grounded in reality. An evident illustration of this phenomenon is presented in the first row of Figure 1. Despite the prompt suggesting a man carrying a bicycle, the LLM-assisted LayoutLLM-T2I tends to overanalyze and assume an additional man riding the bike instead, resulting in the generation of human-like artifacts riding the bicycle, which is not the intended interaction. Such deficiency underscores the critical need for advanced approaches to interpret information from the concise HOI prompts better, ensuring image generation that closely aligns with the intended interactions.

In this paper, we propose Reasoning and Correcting Diffusion (ReCorD) for generating human-object interaction. We argue that proper HOI generation requires both the correct human posture and precise object positioning to facilitate realistic interaction. To achieve this, we introduce an innovative pipeline depicted in Figure 2. Our approach includes three key steps. First, we employ the Latent Diffusion Model (LDM) to produce a set of human candidates performing the verb in the text prompt, emphasizing the correct posture by employing intransitive prompts. Next, as Visual Language Models (VLMs) excel at comprehending image contents, we harness their visual reasoning capabilities to select candidates with optimal posture and determine the appropriate placement of the object based on their interpretation of the interaction scene. Finally, we introduce a refinement mechanism to adjust object positions while preserving accurate human posture. By implementing inverse attention masks and bounding box constraints, we prevent the overlap of attention maps between humans and objects during image generation, enhancing the fidelity of the final image output. Our proposed ReCorD guarantees precise control over the depicted interactions, effectively mitigating the risks of hallucinatory inaccuracies. To the best of our knowledge, this work is the first attempt to introduce interaction generation in a training-free fashion which eliminates the need for additional HOI-labeled data and avoiding computational challenges associated with training. We summarize our main contributions as follows:

(1) We introduce a novel reasoning framework that integrates LDM with VLMs to overcome the challenges of generating realistic HOI, mitigating issues presented in previous approaches, such as LLMs overanalyzing simple text prompts and training data biases in LDM.

(2) To enhance human figure depiction accuracy, we design a correction mechanism within LDM for dynamic image adjustment, enabling precise control and refinement of human interactions in generated images as well as enhancing the portrayal accuracy significantly.

(3) The extensive experiments demonstrate our training-free ReCorD's proficiency in creating captivating and realistic HOI scenes, outperforming state-of-the-art techniques.

## 2 RELATED WORK

### 2.1 Conditioned T2I Diffusion Models

Recent advancements in diffusion models [9, 11, 21, 25, 36, 52] have significantly improved the capabilities of large-scale T2I generation models such as DALL-E [45, 46], Imagen [49], and Stable Diffusion [42, 48]. While these models guided solely by plain text demonstrate the ability to generate high-quality images, they struggle with prompts that demand detailed attribute specifications and a nuanced understanding of spatial relationships [47]. Consequently, recent research on diffusion models extended beyond text-based conditions and incorporated advanced conditioning mechanisms such as inpainting masks [27], sketches [56], key points [29], segmentation maps [10] and layouts [16], facilitating enhanced spatial manipulations. The modification of models marks this evolution to include additional encoders, achieved through strategies like fine-tuning [3, 29, 39, 65], or constructing new models from scratch [24]. For instance, SpaText [3] and GLIGEN [29] introduce spatial modulations into pre-trained models, employing fine-tuning with adapters to perform layout constraints. Despite the enhancements, the requirement for model retraining for each new condition type remains a significant challenge. Building on the foundation of the previous diffusion models, we propose a training-free framework designed explicitly to enhance the proficiency of LDM in interpreting and visualizing the intricate relationships of HOIs.

### 2.2 Image Generation with Spatial Control

As T2I models are traditionally trained on datasets featuring brief text captions, they frequently struggle to capture the nuances of more complex captions that contain multiple phrases [51]. Following the insight from Prompt-to-Prompt [18] that trained T2I models inherently provide token-region associations through their attention maps, several works [2, 4, 7, 16, 22, 27, 59] have been proposed to mitigate this issue. For instance, MultiDiffusion [4] choose to conduct individual denoising procedures for every phrase, respectively, at every timestep. However, this independent generation technique frequently stumbles over lifelike compositions and is easily hindered by bias toward specific actions or objects. Attend-and-Excite [7] strategically manipulates the noise map to enhance the activation of previously overlooked tokens in cross-attention maps. Yet, a limitation arises since the mere intensification of attention to certain tokens does not always lead to a holistic representation of the intended information within the generated output. BoxDiff [59] comes up with three spatial constraints to optimize the cross-attention layers given instinctive inputs, *e.g.*, bounding box or scribble, during the sampling process. InteractDiffusion [22] introduces a novel approach to generating images with precise HOI by tokenizing interaction using a conditioning self-attention layer for the accuracy of the complexities of interaction representation. To better impose regulations on the generated human, the rectangular-shaped constraints cannot be applied since bounding boxes cannot properly convey the details of human action, such as body orientation, facial direction, *etc.* Building on these training-free approaches, our pipeline assures that the object is matched with a sufficiently strong attention map and the human can be generated in a legitimate pose. Thus, a complex scene containing interaction can be precisely rendered.

### 2.3 LLM-assisted Image Generation

The integration of Large Language Models (LLMs) with diffusion models has significantly transformed T2I generation, capitalizing on the superior generalization abilities of LLMs [14, 30, 31, 58, 60, 66]. LayoutGPT [14] adopt LLMs for layout generation via in-context learning. VisorGPT [60] takes one step further by fine-tuning to embrace diverse modalities, including key points, semantic masks, *etc*. LMD [30] represents pioneering efforts in this integration, utilizing LLMs to interpret object locations from text prompts, thus enhancing the accuracy and quality of generated images. LayoutLLM-T2I [43] query ChatGPT for text-to-layout induction and introduce a Layout-aware Spatial transformer with a view to improve layout and image generation simultaneously. Emphasizing the central role of LLMs, these advancements bypass the need for additional information inputs, allowing LLMs to shape the initial layout configurations and interpret user prompts directly. Despite the successful outcomes of using LLMs, they fail to address the challenge of generating the explicit posture of humans given specific actions. Compared to LLMs, LDMs demonstrate a better understanding of scene kinetics based on simple text prompts. Therefore, we propose to leverage such advantage of LDMs alongside the robust visual reasoning abilities of VLMs to generate accurate HOI.

## 3 METHOD

We introduce the ReCorD, an interaction-aware model that maintains training-free superiority. The resulting images hinge on the human adopting the appropriate pose and ensure the object is located in a suitable position according to the given text prompt. Our generative pipeline comprises three modules: Coarse Candidates Generation, Interaction-aware Reasoning, and Interaction Correcting. To abbreviate these modules, we term them as $\mathcal{M}_g$, $\mathcal{M}_r$, and $\mathcal{M}_c$, respectively. We decomposed the denoising process $T$ into two stages, *i.e.* $T_1$ and $T_2$, by observing that the diffusion model captures the initial layout during the early denoising steps and refines the details in the later iterations [4]. In the former stage, $\mathcal{M}_g$ generates $k$ coarse candidates, while $\mathcal{M}_r$ suggests the ideal pose and layout w.r.t. the text prompt. Subsequently, $\mathcal{M}_c$ corrects object locations while preserving selected poses to refine the cursory images into desired ones. Importantly, ReCorD empowers the diffusion model to create images aligned with text prompts, highlighting complex spatial conditions and intricate interactions without additional training. The overall pipeline is depicted in Figure 2, and we elaborate on the details of each module in the following sections.

### 3.1 Coarse Candidates Generation Module

Given a text prompt $y$ containing the HOI attribute, *i.e.*, a statement describes an interaction between a single human and an object phrased as "a subject is verbing an object", we enhance interaction representation by adopting distinct attention mechanisms [55] within $\mathcal{M}_g$. More precisely, we manipulate cross-attention and self-attention maps to generate candidate images $k$ associated with the action subject to the prompt.

**Cross-Attention Maps Manipulation.** To facilitate image generation concerning the textual information, we incorporate such conditions into LDMs using cross-attention maps. During denoising,

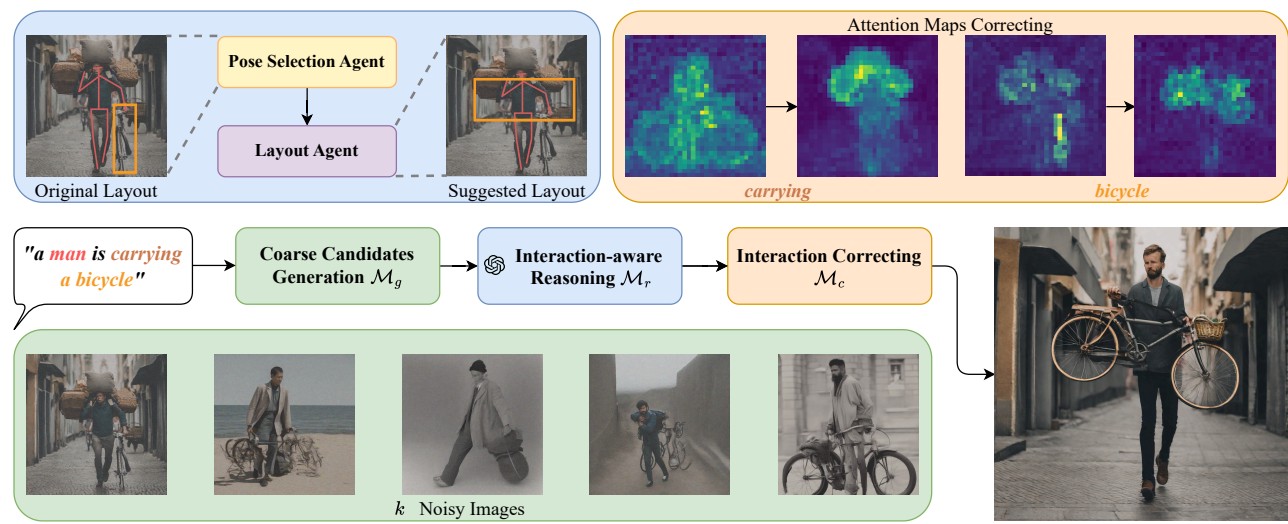

**Figure 2: Overall architecture of ReCorD. Given a text prompt, ReCorD is structured by three components during the inference of LDM and VLMs, where we first leverage Coarse Candidates Generation $\mathcal{M}_g$ to produce coarse candidates. Then, Interaction-aware Reasoning $\mathcal{M}_r$ determines the optimal pose and layout regarding to the input. Finally, Interaction Correcting $\mathcal{M}_c$ adjusts object placements and maintains the chosen poses to enhance the preliminary images within one generation cycle.**

LDMs initially sample a latent vector $z_t$ from a Gaussian distribution $\mathcal{N}(0,1)$ and progressively remove noise to obtain $z_{t-1}$ at each step $t \in [T_1, \cdots, 0]$. After encoding the prompt $y$ into text tokens via text encoder, the cross-attention map is defined as follows:

$$\mathcal{A} = \text{Softmax}\left(\mathbf{QK}^\mathsf{T} / \sqrt{d}\right), \qquad (1)$$

where $\mathbf{Q} = \phi_q(\varphi(z))$ and $\mathbf{K} = \phi_k(\psi(y))$ represent the query and key embeddings derived by corresponding projection functions. $\varphi$ and $\psi$ are spatial normalization [68] and the text encoder from CLIP [44], which yields intermediate representations and $N$ text tokens $\psi(y) = \{w_1 \cdots w_N\}$. For simplicity, we omit the subscript $t$ that represents the denoising step while manipulating attention maps.

Modeling the cross-attention map for the interaction (verbing in the $y$) is challenging when generating HOI scenes, leading to an ambiguous representation of the verb token. To address this issue, we propose an alternative intransitive prompt $\tilde{y}$, which typically excludes object-related descriptions in $y$. The cross-attention maps $\tilde{\mathcal{A}}$ using $\tilde{y}$ can be derived by substituting $\mathbf{K}$ in eq. (1) with $\tilde{\mathbf{K}} = \phi_k(\psi(\tilde{y}))$. As illustrated in Figure 3, $\tilde{\mathcal{A}}$ captures more informative clues, especially for the verb token, compared to $\mathcal{A}$ when using $\tilde{y}$ as the text prompt, resulting in interactive representations. Consequently, we formulate the final cross-attention maps by rearranging the maps as follows:

$$\mathcal{A}_{cross} = \begin{cases} \tilde{\mathcal{A}}_n & \text{if } w_n \in \psi(\tilde{y}) \\ \mathcal{A}_n & \text{otherwise,} \end{cases} \qquad (2)$$

where $n$ denotes the index of text tokens. Ideally, we embrace the attention maps if the text token exists in the intransitive prompt.
**Self-Attention Maps Manipulation.** In contrast to cross-attention maps, self-attention maps lack direct token associations but still influence the spatial layout and appearance of generated images [33].

Therefore, we manipulate the self-attention maps similarly to eq. (2) for latent representation once the denoising step $t > \gamma$ for obtaining $\mathcal{A}_{self}$, where $\gamma$ is a predefined parameter ensuring that scenes and objects from original tokens $\psi(y)$ can be generated effectively.

## 3.2 Interaction-Aware Reasoning Module

As an intermediate module bridging the others, we present the Interaction-Aware Reasoning module $\mathcal{M}_r$ (see Figure 4) following the generation of coarse candidates in $\mathcal{M}_g$. This module comprises two components powered by VLM: the Pose Selection Agent and the Layout Agent. Specifically, the Pose Selection Agent selects an image aligning with the prompt $y$, while the Layout Agent adjusts the object's location and preserves human key points $\mathcal{P}$ and further determines the target position $\hat{b}_o$ for the correction module $\mathcal{M}_c$.
**Pose Selection Agent.** As the pose plays a vital characteristic in the HOI generation, we first couple an agent to select the appropriate pose conditioned on the prompt. The Pose Selection Agent integrates the initial prompt $y$ with previously generated candidates to create the pose template. Leveraging the visual comprehension capabilities of VLMs, this agent excels in identifying the precise pose corresponding to $y$, enhancing the model's ability to interpret visual data beyond relying solely on textual cognition as in LLMs. This pivotal step ensures that the pose information initially obtained from LDMs is meticulously refined for subsequent phases.
**Layout Agent.** To address the issue of LLM-assisted methods being overly reliant on prompts for sampling layouts, we incorporate the identified key points $\mathcal{P}$ and the bounding box $b_h$ for humans as additional data. Recognizing that an interaction involves both the relation to human and the object, we collect the crucial information of $\mathcal{P}$ using 33 key points in $(x, y)$ coordinates and represent $b_h = (x_{min}, y_{min}, x_{max}, y_{max})$. Additionally, we use the image selected by the former agent as inputs to VLMs for layout suggestion tasks.

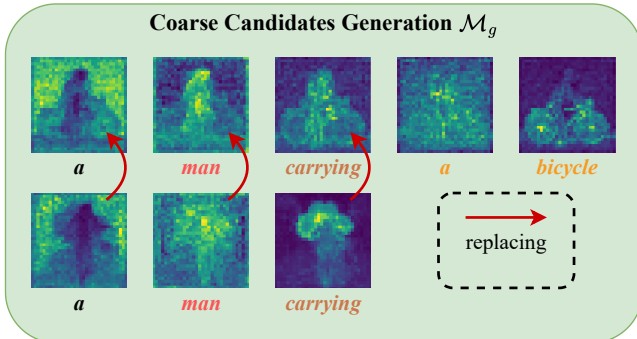

**Figure 3: For a text prompt involving HOI, $\mathcal{M}_g$ attempts to generate coarse candidates by substituting the attention maps obtained from the full prompt (top row) with those derived from the intransitive prompt (bottom row).**

We first extract the object's location $b_o$ using Otsu's algorithm [41], an automatic thresholding technique applied to the object's cross-attention map $\mathcal{A}_{cross}$ to isolate regions with higher values. Subsequently, we detect human key points using MediaPipe Pose Landmarker to create the segmentation mask $m_h$. Consecutively, we establish a series of guidelines and fixed protocols for VLMs adherence, including constraints on $b_h$ to maintain the integrity of the intended human poses and an overlapping reduction strategy to improve the quality of generated images containing multiple objects. Furthermore, inspired by the Chain-of-Thought approach [57], we enhance the logical coherence by guiding VLMs to construct visual attribute information for human posture. We ground the VLMs in logical reasoning across multiple factors such as pose types, body orientation, object relations, *etc.* Drawing on insights from previous research [5, 37], we prepare VLMs with three examples, aiding in the clarification of visual representation and reducing ambiguity to construct the interaction template. Eventually, we extract the proposed location $\hat{b}_o$ for $\mathcal{M}_c$ and integrate a checking mechanism to determine whether the alteration in $\hat{b}_o$ falls within a predetermined threshold. If the change is minimal which indicates a minor difference, $\mathcal{M}_r$ would signal *no changes* to $\mathcal{M}_c$. This mechanism is crucial for maintaining a streamlined and resource-efficient generation process, ensuring only significant location adjustments prompt further action. Please refer to the supplementary material for the details of pose and interaction templates.

### 3.3 Interaction Correcting Module

We delicately refine the candidate image provided by the dual agents while preserving the original human pose in $\mathcal{M}_c$, as shown in Figure 5. To combine the generative capabilities of LDMs with the reasoning abilities of VLMs, we incrementally update the latent $z_t$ to adjust the object's position and size based on bounding boxes $\hat{b}_o$ related to the interaction. Notably, we conduct the denoising process for $t \in [T_2, \cdots, 0]$, including modulation of cross-attention and self-attention maps, as described in Section 3.1.

Simultaneously modifying the object's location requires consideration of potential overlap with the human body since cross-attention maps from different tokens may exhibit strong values in

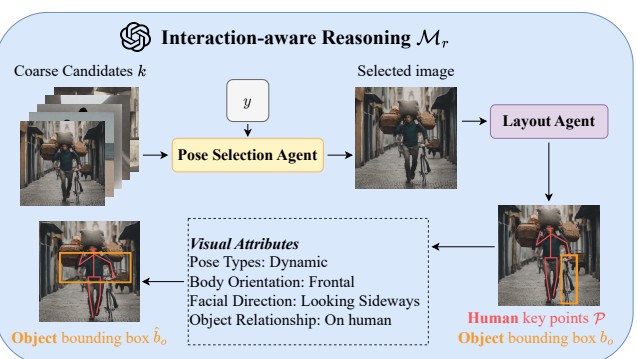

**Figure 4: Given $k$ coarse images, the Pose Selection Agent identifies the image most closely aligning with the text prompt $y$, and the Layout Agent updates the object's position $\hat{b}_o$ by reasoning arrangements while preserving the pose $\mathcal{P}$.**

the same region which will deteriorate image quality. To address this challenge, we introduce a mechanism for eliminating attention overlap. Specifically, given the token index of the object denoted as $m$, we use the cross-attention map $A_m$ to construct an inverse mask at each time step $t$, denoted as $\bar{A}_m = \mathbb{1} - A_m$, where $\mathbb{1}$ is the tensor of the same dimension as $A_m$ containing all elements equal to 1. This inverse attention map is then applied to the remaining maps using an element-wise product operation defined as

$$\hat{A}_n = \bar{A}_m \odot A_n, \quad \forall n \neq m. \tag{3}$$

Through this correction mechanism outlined in eq. (3), we can mitigate the issue of attention overlap between humans and objects while updating the object positions, ensuring the successful generation of updated objects.

**Conditioned Spatial Constraints** Since our ReCorD is training-free and does not involve additional learnable networks for knowledge transfer, we employ box constraints [59] to regularize the denoiser, which can be formulated as

$$\mathcal{L} = \mathcal{L}_{IB} + \mathcal{L}_{OB} + \mathcal{L}_{CC}, \tag{4}$$

where each term in sequence order represents the inner-box, outer-box, and corner constraint, respectively. We apply eq. (4) to update the latent at each time step $t$ with corresponding weight $\alpha_t$ as

$$\hat{z}_t \leftarrow z_t - \alpha_t \cdot \nabla \mathcal{L}. \tag{5}$$

Through slight update $z_t$ at each step, we ensure that the object holds sufficient mutual information with the box region and conforms to the specified size, *i.e.* $\hat{b}_o$, thereby accurately correcting the object's position to represent the interaction. As LDM aims to denoise iteratively and involve the attention maps as intermediate. We denote that LDM($z_t, y, t, s$) is the diffusion process at time step $t$ before manipulation, which seeks the corresponding attention maps. After going through our proposed ReCorD, the denoising UNet is reused to predict the latent representation at next step. Formally LDM($\hat{z}_t, y, t, s$) symbolizes the diffusion model that adopts the manipulated attention maps, resulting in the prediction, *i.e.* $z_{t-1}$. The complete algorithm can be found in the supplementary.

**Table 1: Comparison between ReCorD and existing baselines in terms of generated image quality scores in $\mathcal{S}_{\text{CLIP}}$, $\mathcal{S}_{\text{CLIP}}^{verb}$, PickScore, FID, along with HOI classification score on HICO-DET and VCOCO. Ours$^\dagger$ represents using SDXL as backbone.**

| Method | HICO-DET | | | | | | V-COCO | | | | |
| --- | --- | --- | --- | --- | --- | --- | --- | --- | --- | --- | --- |
| | $\mathcal{S}_{\text{CLIP}}\uparrow$ | $\mathcal{S}_{\text{CLIP}}^{verb}\uparrow$ | PickScore $\uparrow$ | FID $\downarrow$ | HOI$_{\text{Full}}\uparrow$ | HOI$_{\text{Rare}}\uparrow$ | $\mathcal{S}_{\text{CLIP}}\uparrow$ | $\mathcal{S}_{\text{CLIP}}^{verb}\uparrow$ | PickScore $\uparrow$ | FID $\downarrow$ | HOI $\uparrow$ |
| SD [48] | 31.74 | 21.82 | 21.50 | 51.31 | 18.78 | 10.02 | 31.10 | 21.26 | 21.26 | 77.29 | 15.85 |
| A&E [7] | 31.63 | 21.72 | 21.33 | 46.41 | 16.57 | 8.62 | 31.21 | 21.11 | 21.11 | 70.74 | 14.52 |
| LayoutLLM-T2I [43] | 31.63 | 22.02 | 21.01 | 38.94 | 16.98 | 8.06 | 31.65 | 21.62 | 20.88 | 59.35 | 16.64 |
| BoxDiff [59] | 31.42 | 21.69 | 21.22 | 45.88 | 16.33 | 8.67 | 31.06 | 21.27 | 20.96 | 68.67 | 12.34 |
| InteractDiffusion [22] | 28.72 | 21.34 | 20.40 | **29.74** | 21.57 | 10.25 | 28.34 | 20.76 | 20.16 | **49.74** | 15.78 |
| MultiDiffusion [4] | 31.64 | 21.81 | 21.67 | 51.51 | 22.46 | 11.15 | **32.53** | 21.31 | 21.81 | 83.27 | 17.96 |
| SDXL [42] | 32.06 | 22.29 | **22.68** | 40.32 | 25.85 | 14.24 | 31.76 | 21.40 | **22.54** | 75.40 | 19.02 |
| LMD [30] | 28.67 | 20.11 | 20.62 | 51.37 | 9.10 | 2.65 | 29.31 | 20.29 | 20.56 | 75.68 | 10.26 |
| Ours | 31.92 | 22.26 | 21.49 | 37.03 | 22.86 | 12.72 | 31.60 | 21.55 | 21.31 | 58.20 | 20.00 |
| Ours$^\dagger$ | **32.40** | **22.65** | 22.54 | 36.72 | **26.33** | **15.39** | 31.94 | **21.84** | 22.22 | 60.74 | **22.48** |

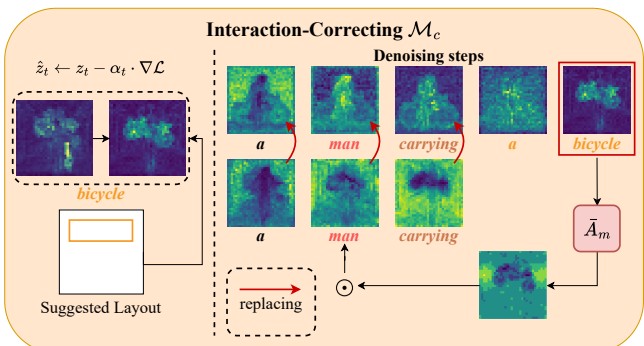

**Figure 5: $\mathcal{M}_c$ refine the coarse candidate according to the suggested layout by adjusting object location and size based on Eq. 4, employing the inverse mask $\bar{A}_m$ along with an element-wise product to deal with the attention overlapping concerns.**

## 4 EXPERIMENTS

### 4.1 Experimental Setup

**Datasets.** Given the absence of a standard benchmark crafted for HOI generation, we assess the efficacy of our approach by extracting HOI triplets from two established HOI detection datasets, namely HICO-DET [6] and VCOCO [17], to form the input text prompts. HICO-DET includes 600 triplets across 80 object categories and 117 verb classes, while VCOCO contains 228 triplets, spanning 80 object classes and 29 verb types. For a comprehensive assessment, we incorporate the non-spatial relationship category in T2I-CompBench [23], which is characterized by 875 interaction terms. We select prompts that exclusively involve HOIs in T2I-CompBench. To enhance diversity, we apply random subject augmentation to each verb and object pair extracted from the datasets to form the input prompt. Accordingly, our experiments are conducted across three datasets: HICO-DET, with 7,650 HOI prompts; VCOCO, contributing 2,550 prompts; and the non-spatial relationship category of T2I-CompBench, adding 465 prompts. More details are described in the supplementary.

**Baselines.** We report comparisons with nine strong-performing models, 1) T2I models: Stable Diffusion (SD) [48], Attend-and-Excite (A&E) [7], SDXL [42], and DALL-E 3. 2) L2I models: BoxDiff [59], MultiDiffusion [4] and InteractDiffusion [22]. 3) LLM-assisted T2I models: LayoutLLM-T2I [43] and LMD [30]. We utilized the official implementations and the default settings for each baseline. For L2I models, we provide the actual bounding box data from HICO-DET and VCOCO datasets in addition to the text prompts as inputs. For LLM-assisted methods, the input layouts are exclusively generated by LLMs, rather than being sourced from the datasets.

**Evaluation Metrics.** To measure the interaction in the generated images, we utilize the CLIP-Score $\mathcal{S}_{\text{CLIP}}$ [19] evaluating the similarity between the input text and the generated images. While this metric is commonly applied to estimate fidelity to text prompts, we note its inclination towards a noun or object bias, with CLIP often unable to differentiate among verbs, relying instead on nouns [38, 64]. To address this, we specifically extract verbs from the text prompts and calculate the Verb CLIP-Score $\mathcal{S}_{\text{CLIP}}^{verb}$. In addition, we introduce a HOI classification score to evaluate the interaction depiction. By transforming a pre-trained, SOTA HOI detector [67] into a classifier, we evaluate HOI instances in generated images and compare them against the ground truth of HICO-DET and VCOCO. The accuracy of HOI classification is evaluated based on the top three accuracy scores. HOI$_{\text{Full}}$ and HOI$_{\text{Rare}}$ represent scores for the full and rare set, respectively, on the HICO-DET dataset. The rare set is selected based on having fewer than 10 instances across the dataset. Moreover, we employ the Fréchet Inception Distance (FID) [20] and PickScore [28] to assess image quality. FID compares the Fréchet distance distribution of Inception features between real and generated images, whereas PickScore, a text-image scoring metric, exceeds human performance in predicting user preferences.

**Implementation Details.** We choose the Stable Diffusion [48] model as the default backbone and GPT4V [1] as the VLM in $\mathcal{M}_r$. We set the ratio of classifier-free guidance to 7.5, denoising steps $T_1 = 10$, $T_2 = 50$, and use the DDIM [52] scheduler within denoising steps. The number of coarse candidates $k = 5$, and the hyperparameter $\gamma = 5$ initials the operation for self-attention maps manipulation. For evaluation, one image is generated per triplet for HICO-DET and VCOCO datasets and three images per triplet for T2I-CompBench.

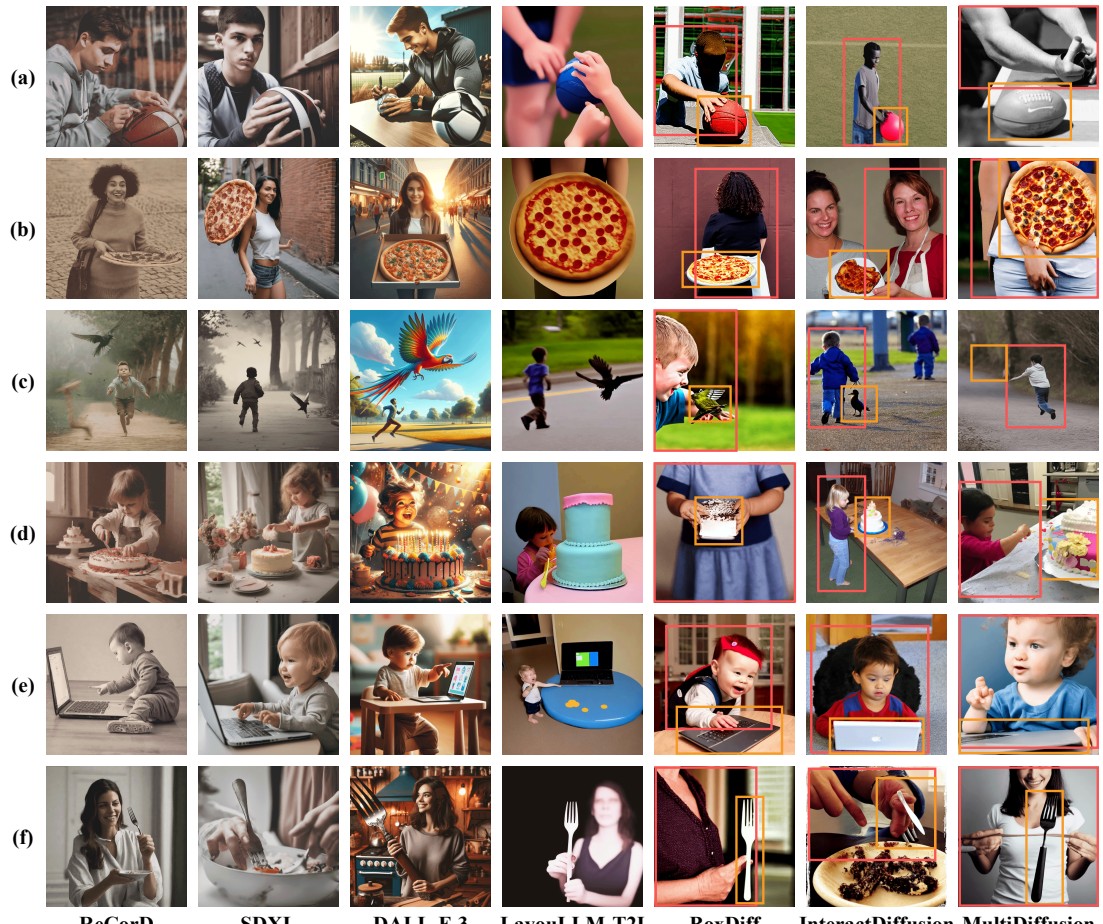

**Figure 6: Visual comparison with existing baselines for HICO-DET (a-c) and VCOCO (d-f) using different text prompts, where ReCorD attains better delineation of interaction, and renders images matching the text instructions. (a) a young man is signing a sports ball. (b) a woman is carrying a pizza. (c) a boy is chasing a bird. (d) a child is cutting a cake. (e) a toddler is pointing at a laptop. (f) a woman is holding a fork. The bounding boxes on the results of L2I models are additional input for HOI generation.**

## 4.2 Qualitative Results

We provide a qualitative comparison to assess the generated HOIs. As depicted in Figure 6, ReCorD outperforms other SOTA methods by generating realistic human poses and object placements that align with text prompts, proofing its proficiency in depicting object interactions with high fidelity. In contrast, baseline methods often tend to misplace objects or fail to capture the nuances of intended actions. For L2I models with additional layout inputs, BoxDiff achieves object size requirements but struggles to accurately depict interaction poses; InteractDiffusion fails to accurately portray subtle activities despite the fine-tuning, as seen in (a), (d), (e), and (f); MultiDiffusion strives for precise object placement despite generating images in various sizes. On the other hand, LayoutLLM-T2I, despite leveraging language models for improved layout generation, often produces objects disproportionate to humans, which is evident in (e) and (f). Furthermore, the tendency of the refiner module in SDXL as well as LMD to prioritize nouns over verbs compromises their interaction depiction capabilities. Especially, SDXL struggles

with action poses (a), (b), (d), and (e), and DALL-E 3 with object sizing and placement (a), (c), (e), and (f), illustrating key areas where ReCorD advances beyond the limitations of existing solutions.

## 4.3 Quantitative Results

We present the quantitative comparison of the generated results, with prompts sourced from HICO-DET and VCOCO in Table 1, and results using prompts formed from T2I CompBench in Table 2.

**CLIP-based Image-Text Similarity.** The results of CLIP-Score $\mathcal{S}_{\text{CLIP}}$ reveal that our ReCorD outperforms other methods on HICO-DET and T2I-CompBench, and it is comparable to MultiDiffusion on VCOCO. Furthermore, our ReCorD achieves the best results across all three datasets in terms of Verb CLIP-Score $\mathcal{S}_{\text{CLIP}}^{verb}$, this confirms our ability to generate more closely matched interactions.

**Evaluation of Image Quality.** As per PickScore evaluation, our ReCorD model is comparable with the SDXL model and outperforms other methods. This demonstrates that after incorporating our designed *Interaction-Correcting Module* with SD models. Our

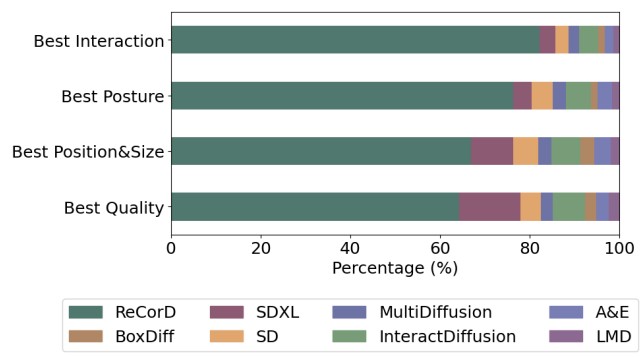

Figure 7: User Study on HOI Generation

ReCorD can maintain the image generation quality of the model while achieving more authentic interaction. Furthermore, when comparing the generated images with real ones in HICO-DET and VCOCO datasets using FID scores, our model outperforms other methods except InteractDiffusion. Notably, given that InteractDiffusion is fine-tuned using HICO-DET and COCO datasets [32], the performance of our ReCorD is particularly remarkable as it operates without the need for training or additional HOI data.
**Evaluation of Interaction Accuracy.** Table 1 validates our method significantly enhances the accuracy of HOI generation in both datasets, revealing the efficacy in synthesizing more precise HOI.

### 4.4 User Study

We conduct a user survey with 95 participants who are asked to evaluate images generated by baseline methods and our ReCorD based on four criteria: *Interaction, Posture, Position&Size, and Overall Quality*. We generated eight distinct images for each of 24 randomly selected prompts from the HICO-DET and VCOCO datasets, using methods including A&E, LMD, SD, SDXL, MultiDiffusion, InteractDiffusion, BoxDiff, and ReCorD. According to the results represented in Figure 7, ReCorD emerged as the clear favorite, garnering between 60% to 80% of the votes across the criteria, indicating its effectiveness in accurately generating the correct postures and positioning objects within reasonable areas, which are key factors ensuring the accurate depiction of interactions. The results of the user study illustrate that our ReCorD can better align with human understanding of HOIs, underscoring its advanced capability to not only recognize but also precisely delineate the nuances of HOIs.

### 4.5 Ablation Studies

For the ablation studies, Figure 8 displays the HOI generation results of the SD incorporating modules in our ReCorD : (a) with only SD, (b) with SD and $\mathcal{M}_g$, and (c) with SD and $\mathcal{M}_g + \mathcal{M}_r + \mathcal{M}_c$. With only SD, the generated outputs appear to be suboptimal, likely influenced by biases inherent in its training data, leading to misinterpretation of the intended interaction described in the text prompts. From the results in (b), it shows that with the inclusion of $\mathcal{M}_g$, the accuracy of the generated human action is significantly improved due to our intransitive prompt altering technique, highlighting that simplifying the prompt to focus on the core action enables the model to generate the intended human poses with enhanced precision.

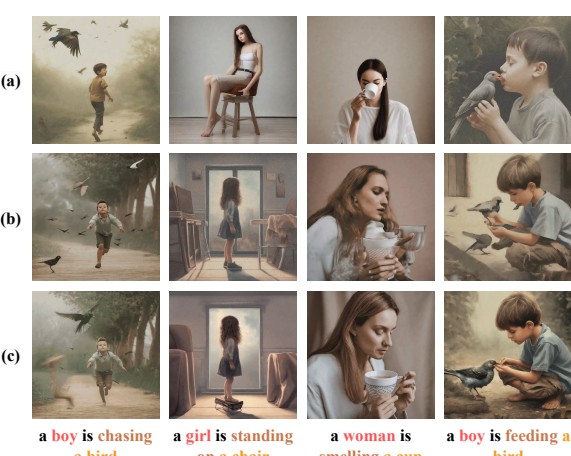

a **boy** is **chasing a bird**  a **girl** is **standing on a chair**  a **woman** is **smelling a cup**  a **boy** is **feeding a bird**

Figure 8: Ablation study of integrating different modules: (a) SD, (b) SD +$\mathcal{M}_g$, (c) SD + $\mathcal{M}_g + \mathcal{M}_r + \mathcal{M}_c$. Evidently, (a) depicts biased actions, (b) shows factual actions yet with flawed interaction, and (c) generates genuine interaction images with proper object locations while keeping the correct actions.

However, it still struggles to accurately position objects in relation to humans in the images, resulting in a mistaken interaction. For our ReCorD, $\mathcal{M}_r$ helps select appropriate poses and retain suitable candidates while $\mathcal{M}_c$ refines images to obtain accurate interaction with the correct object size and position while maintaining the selected pose. As a result, HOI generation of our complete pipeline in (c) exemplifies the most successful outcomes achieved.

Table 2: Comparison with SOTAs on T2I-CompBench.

| Method | $\mathcal{S}_{\text{CLIP}} \uparrow$ | $\mathcal{S}_{\text{CLIP}}^{verb} \uparrow$ | PickScore $\uparrow$ |
|---|---|---|---|
| SD [48] | 30.03 | 21.39 | 20.96 |
| A&E [7] | 29.59 | 21.65 | 20.33 |
| LayoutLLM-T2I [43] | 30.35 | 22.13 | 20.36 |
| MultiDiffusion [4] | 30.59 | 21.74 | 21.14 |
| SDXL [42] | 30.44 | 21.86 | **21.82** |
| LMD [30] | 27.27 | 20.63 | 19.94 |
| Ours | 30.14 | 21.94 | 20.83 |
| Ours$^\dagger$ | **30.71** | **22.38** | 21.64 |

## 5 CONCLUSION

We have introduced ReCorD framework, tailored explicitly for HOI image generation. This method comprises three interaction-specific modules that synergistically interact with each other. Our core idea revolves around reasoning layout and correcting attention maps using VLM-based agents and an LDM to address this challenge. Extensive experiments demonstrate the effectiveness of our approach in enhancing image accuracy and semantic fidelity to the input text prompts, particularly in capturing intricate concepts of interactions that several baseline generative models struggle with. Additionally, we quantify our improvements through various protocols and a user survey focused on HOI generation, providing valuable insights and paving the way for future explorations in this domain.

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
