# OpenReview forum: "ReCorD: Reasoning and Correcting Diffusion for HOI Generation"
_acmmm.org/ACMMM/2024/Conference — MM2024 Poster_

### Official Review · Reviewer_WhNq · 2024-05-12

**Rating:** 4
**Confidence:** 2

**Summary:**

The paper focuses on the inaccurate representation of prompt content in existing diffusion model-based generative models. It proposes a paradigm based on the decoupling and reconstruction of internal relationships within human, object, and interaction triplets. This is achieved by introducing the reasoning-capable LDM and VLM interaction to determine the optimal pose and layout represented by the prompt and provide guidance in the new generation process. Multiple indicators have proven the effectiveness of the approach.

**Strengths:**

1. The paper is well-written, allowing readers to quickly understand the authors' ideas and focus.

2. The focus of the paper is meaningful because human, object, and interaction are the most common elements found in prompt texts, and a correct understanding of their relationship is essential.

3. The idea of going from coarse to fine is reasonable as it allows for minimal changes to the original layout of the image, preventing overall layout changes caused by modifications to objects or interactions.

4. The design of the Interaction-Aware Reasoning Module has a certain level of reliability and can further eliminate ambiguity.

5. The visualization results are quite impressive, surpassing the image quality of some fine-tuning based approaches to a great extent.

**Limitations:**

1. The introduction of VLM and multiple noisy images requires further discussion on time and resource costs.

2. The process of Attention Maps Correcting can be further refined.

3. The guiding role of suggest Layout throughout the denoising time steps can be further explored.

4. For instructions used to generate images, it is necessary to consider cases with more factors in the prompt, such as the appearance of image styles, the presence of animals as subjects, new backgrounds, and other scenarios.

**Suitability:**

2

---

### Official Review · Reviewer_eRxq · 2024-05-12

**Rating:** 4
**Confidence:** 3

**Summary:**

The *ReCorD* proposed in the paper is to make HOI generation more reasonable. LLM selects the most appropriate pose from the coarse noisy images generated with object-related descriptions. Using the layout suggested by LLM, attention maps are corrected, after which the HOI image is synthesized. Fruitful experiments show the capability of the framework in HOI generation objectively.

**Strengths:**

1. A novel framework for reasoning HOI generation is proposed in the paper. The reasoning layout is suggested by LLM with the heuristic of input candidates, which simultaneously enhances the efficiency of the LLM rather than directly generating it.

2. The paper offers a framework for LLMs to operate the attention map during the procedure of LDMs, which is remarkable. The framework generally outperforms the others in metrics ranging from CLIP-Score to FID.

3. A specific dataset is constructed for HOI evaluation and extensive ablation studies have been implemented in the paper, which demonstrates the efficiency of the *ReCorD*.

4. The paper is clearly written and is easy to follow. Good paper. Source codes are provided in supplementary material which is commendable.

**Limitations:**

1. I find that from the results shown in the paper, the images generated by *ReCorD* always have a lower brightness than the others. However, this result is from the previous research where token-level amending results in less fruitful color. The discussion of this phenomenon would increase your contributions and innovations.

2. More ablations could be implemented to compare the ability of different LLMs or MLLMs in layout suggestions.

3. GPT4-V can be prompted with designed heuristics to play the role of the HOI generation evaluator.

**Suitability:**

3

---

### Official Review · Reviewer_LxrP · 2024-05-26

**Rating:** 4
**Confidence:** 3

**Summary:**

This paper introduces a framework, ReCorD, aimed at generating Human-Object Interaction (HOI) images. The primary innovation of this framework lies in its integration of the Latent Diffusion Model (LDM) with the Visual Language Model (VLM) to handle complex spatial conditions and intricate interactions without additional training. The LDM generates a set of human figures as coarse candidates. The VLM selects candidates with optimal postures and determines the precise placement of objects within the scene. The correcting module adjusts object positions accurately while maintaining the integrity of human postures. Both qualitative and quantitative experiments demonstrate the framework's superiority over existing methods in certain aspects.

**Strengths:**

1. The authors innovatively integrate the VLM and LDM for generating realistic HOI. With object bounding box refinement, it enables precise control and refinement of human interactions in generated images. No training is required, which enhances its usability and scalability.
2. The experimental design is relatively comprehensive, with a selection of quantitative metrics, multiple visual comparisons showcased through various case studies, and user studies conducted, validating the effectiveness of the method from multiple perspectives.
3. The article is well-written, with clear descriptions and aesthetically pleasing and informative visual representations.

**Limitations:**

1.	Visual effects are often the most direct way to assess the capabilities of a generative model. However, compared to other methods, the visual effects presented in this paper mostly feature dark images. The authors did not explain it, leaving it unclear if there are specific reasons for this phenomenon. Are there any better examples?
2.	[59] is the key point in the correcting module, thus, the refined object bounding box is very important. The bounding box replacement should be illustrated in detail, and the connection with the visual attributes should be reported.
3.	In the quantitative experiments, the difference between the proposed method and its backbone is subtle. In table 1, “Ours” is better than SD but worse than SDXL. Does this mean the HOI challenge could be handled by upgrading model scales and training data.
4.	The proposed method involves a set of candidate generation, attention map manipulation, auxiliary modules, etc. The total generation speed and memory usage should be reported.

**Suitability:**

3

---

### Official Review · Reviewer_MMAh · 2024-05-31

**Rating:** 4
**Confidence:** 3

**Summary:**

Diffusion models revolutionize image generation by leveraging natural language to guide the creation of multimedia content. Despite significant advancements in such generative models, challenges persist in depicting detailed human-object interactions, especially
regarding pose and object placement accuracy. We introduce a training-free method named Reasoning and Correcting Diffusion (ReCorD) to address these challenges. Our model couples Latent
Diffusion Models with Visual Language Models to refine the generation process, ensuring precise depictions of HOIs. We propose an interaction-aware reasoning module to improve the interpretation of the interaction, along with an interaction correcting module
to refine the output image for more precise HOI generation delicately. Through a meticulous process of pose selection and object positioning, ReCorD achieves superior fidelity in generated images
while efficiently reducing computational requirements.

**Strengths:**

1） a novel reasoning framework that integrates LDM with VLMs to overcome the challenges of generating realistic HOI, mitigating issues presented in previous approaches, such as LLMs overanalyzing simple text prompts and training data biases in LDM.
2） a correction mechanism within LDM for dynamic image adjustment, enabling precise control and refinement of human interactions in generated images as well as enhancing
the portrayal accuracy significantly.

**Limitations:**

1）The image quality scores are not state-of-the-art
2)  The visual effect is not unreal

**Suitability:**

3

---

### Meta-Review · Area_Chair_wrtS · 2024-06-27

**Recommendation:** Accept (Poster)
**Confidence:** 4

**Metareview:**

This paper presents an approach to T2I generation by integrating diffusion models with VL models to tackle the intricate challenge of human-object interaction (HOI). By utilizing LDMs for generating a variety of pose options and VLMs for selecting the optimal poses and object placements based on visual cues and textual descriptions, the proposed method ensures the fidelity and contextual coherence of generated images.

The paper receives the initial rating of four borderline accept. After the rebuttal, two reviewers raise the rating to weak accept (4 4 5 5). The AC agrees with the reviewers' comments that this paper provides interesting insights to HOI generation.